# Antibacterial Activity of Dissolved Silver Fractions Released from Silver-Coated Titanium Dental Implant Abutments: A Study on *Streptococcus mutans* Biofilm Formation

**DOI:** 10.3390/antibiotics12071097

**Published:** 2023-06-24

**Authors:** Ranj Nadhim Salaie, Pakhshan A. Hassan, Zhala Dara Meran, Shehab Ahmed Hamad

**Affiliations:** 1Department of Oral and Maxillofacial Surgery, Faculty of Dentistry, Tishk International University, Erbil 44001, Iraq; 2Department of Biology, College of Science, Salahaddin University, Erbil 44001, Iraq; pakhshan.hassan@su.edu.krd; 3Department of Prosthodontics, College of Dentistry, Hawler Medical University, Erbil 44001, Iraq; zhalla.meran@hmu.edu.krd; 4Kurdistan Higher Council of Medical Specialties, Erbil 44001, Iraq; shehab.ahmed@khcms.edu.krd

**Keywords:** antimicrobial nanoparticles, dental materials, nanocoatings, nanodentistry

## Abstract

(1) Background: The aim of this research was to investigate the antibacterial activity of dissolved silver from silver-coated titanium implants against *Streptococcus mutans*. (2) Methodology: Silver-coated titanium implant discs were immersed in 1.8 mL of brain heart infusion broth (BHIB) and incubated for 24 h in order to release the silver ions into the broth. The coating quality was confirmed via EDS, and the dissolved silver was measured via inductively coupled plasma mass spectrometry (ICP-MS). The experimental design used unconditioned broth (control) and broth conditioned with silver released from silver-coated titanium implants (*n* = 6). Regarding the antibacterial activity, isolated *Streptococcus mutans* was used. A turbidity test and lactate production test were performed to determine the effect of dissolved silver on bacterial growth in a suspension and biofilm formation. (3) Result: The results showed that the coating was successfully applied on the substrate. There was around 0.3 mg/L of silver released into the BHIB, and the turbidity of the control group was significantly higher than the treatment, with measured absorbance values of 1.4 and 0.8, respectively, indicating that the dissolved silver ions from the silver-coated titanium discs exhibited some degree of antibacterial activity by preventing the growth of *Streptococcus mutans*. However, the results of the antibiofilm activity test did not show any significant difference between the groups. (4) Conclusion: The dissolved silver from silver-coated titanium implants has an antibacterial activity but not a significant antimicrobial activity, indicating that the dissolved silver from silver-coated titanium abutments can significantly reduce the incidence of peri-implant mucositis.

## 1. Introduction

Dental implants have long been used as replacements for missing teeth [1]. Titanium (and its alloy) are the most common biomaterials used to manufacture dental implants [2]. Despite their wide use for replacing missing teeth, dental implants still face problems that limit their long-term clinical success. Peri-implantitis is one of the main causes of dental implant failure [2,3]. Bacteria adhere to the implant surfaces via the formation of a complex biofilm which can be considered the main factor in peri-implantitis [4]. The biofilm is a microbial-derived, sessile community characterized by microbes that are irreversibly attached to a substratum and/or interface with each other, embedded in a matrix of extracellular polymeric substances produced by the microbes [5]. Bacteria are widespread in the oral cavity; saliva contains 108 bacteria/mL, along with proteins and glycoproteins that selectively bind to the surfaces of teeth and dental implants [5]. Bacteria can adhere to the supra gingival part of the dental implant (i.e., the abutment and crown), which causes inflammation that consequently becomes peri-implantitis and causes bone loss [6].

Researchers have tried to modify the dental implant surface in an attempt to improve osseointegration [7,8]. However, the development of a modified implant surface or a coating with antibacterial activity in still in progress [9,10]. Researchers developed an antimicrobial coating loaded with antibiotics to prevent bacterial colonization on the implant surface which subsequently leads to infection; however, due to the burst release of the antibiotic, the coating did not exhibit long-term antibacterial activity [11,12]. Furthermore, attempts were made to load the titanium dental implant surface with chlorhexidine [13]. However, there were some reports of the cytotoxic effect of chlorhexidine against human cells [14].

Recently, nanotechnology has broadened the scope of research on improving biomaterials. Nanoparticles are commonly incorporated into dental implants and other biomaterials in the form of coatings [15]. Silver nanoparticles have widely been used in the medical field because of their strong antimicrobial activity [16]. Researchers found that the antimicrobial activity of silver nanoparticles is better than the traditional chlorhexidine disinfectant used in dentistry, reporting that the MIC of dissolved silver from silver particles was 0.09 mg/L [17]. However, Vilarrasa and co-workers found that incorporating nanosilver into the surface of titanium did not provide a significant degree of antibiofilm activity [18]. Moreover, it was found in another study that the minimum concentration of silver needed to inhibit Gram-positive and Gram-negative species was only 0.05 ppm [19]. These different findings indicate that the bioavailability of silver (either as a particle or as a coating) is different according to the experimental conditions, coating procedure, primary particle size, amount of silver released, content and volume of the culture media, and bacterial strain and growth. The exact mechanism of the antibacterial activity of silver nanoparticles is still being elucidated. One of the possibilities is the release (dissolution) of free metal ions from the surface of the particle; another possibility is oxidative stress via the generation of reactive oxygen species (ROS) from the surface of particles [17,20,21]. Nevertheless, it is still unclear whether the toxicity is caused by the particles themselves or from the dissolved ions [22].

Incorporating silver into the surface of titanium dental implants (or dental implant abutments) might significantly decrease the incidence of peri-implant infection by releasing silver ions into the peri-implant environment (or gingival sulcus, if the dental implant abutment is coated with silver). Thus, the aim of this study was to investigate the antibacterial and antibiofilm activity of released (dissolved) silver from silver-coated titanium dental implants (or the dental implant abutment). The antibacterial property of the dissolved silver was tested against *Streptococcus mutans*, which is one of the most common pathogens included in peri-implant infections. The objectives were to successfully coat the titanium discs, measure the release of silver into the BHIB, and finally investigate the antibacterial activity of the released (dissolved) silver against *S. mutans.*

## 2. Results

The results confirm the that the coating was successfully applied to the substrate and that the particles were nano-sized. The coating quality was confirmed via EDS analysis (Figure 1). The release of silver from the silver-coated titanium discs was also confirmed. Regarding the antibacterial activity, it was shown than the dissolved silver significantly decreased bacterial growth. However, the antibiofilm activity of the dissolved silver was not confirmed.

### 2.1. Release of Silver from the Coating

The silver released into the BHIB from the coated implant discs was tested via ICP-MS. The results showed that there was 0.34 ± 0.03 mg/L of silver in the BHIB after 24 h of its incubation with silver-coated titanium discs. However, the silver concentration in the control was less than the detection limit (Figure 2).

### 2.2. Assessment of Bacterial Viability in Suspension (Turbidity Measurement)

The turbidity of the BHIB after its incubation with *S. mutans* was measured as a marker of bacterial growth. The results showed that the turbidity of the BHIB in the treatment group was significantly less than the control group (*p* = 0.023), measuring 0.88 ± 0.19 and 1.46 ± 0.22, respectively (Figure 3).

### 2.3. Antibiofilm Activity

The results of the antibiofilm activity of the dissolved silver showed that the turbidity of the treatment group was greater than the control groups. However, there was no statistically significant difference between them (*p* = 0.14), (Figure 4).

### 2.4. Determination of Lactate Production

Lactate production was tested as an indication of the presence of metabolically active bacteria. The results showed that the lactate production by *S. mutans* in the media (suspended bacteria) was significantly higher in the control compared to the treatment (*p* = 0.018), which measured 10 ± 1.1 and 5 ± 1.2, respectively (Figure 5).

Regarding the lactate production by *Streprococcus mutans* in the biofilm, there was no significant difference in lactate production between the control and treatment (*p* = 0.16), which measured 9 ± 1 and 10 ± 1.5 mm, respectively (Figure 6).

## 3. Discussion

This study investigated the antimicrobial activity of dissolved silver from silver-coated titanium dental implant abutments against *S. mutans*. The main findings confirmed the presence of dissolved silver in the BHIH which exhibited an antibacterial property without having a considerable antibiofilm activity. There was around 0.3 mg/L of dissolved silver in the BHIB, which indicates that electroplated silver nanoparticles on metal substrates dissolve and release silver ions into biological media and possibly blood and gingival crevicular fluid [9]. This finding is supported by another study, which found that the silver released from silver-anchored titanium dioxide nanotubes on titanium implants amounted to 0.5 mg/L after 24 h of incubation in BHIB [23]. This could be explained by the fact that there is a high affinity between silver and the -SH group which is present in amino acids in biological media. Moreover, it has been shown that the presence of organic molecules in the biological media (and blood and gingival fluid) can accelerate the dissolution of silver ions [24]. Nevertheless, Salaie and co-workers found that the silver released from -coated titanium dental implants was around 1 mg/L after 24 h of incubation in osteoblast cell culture media [25]; the same study found that the presence of fetal bovine serum (FBS) in a cell culture media significantly increased silver dissolution. This could be explained by the fact that FBS can significantly enhance the dissolution of metal nanoparticles [26]. Furthermore, it could be argued that the dissolution of silver in biological media depends on many factors. Thus, the amount and bioavailability of silver dissolved from the silver-coated implant abutment into the gingival sulcus (crevicular fluid) might be different from what was found in the current study (BHIB). This is because the content and volume of the BHIB used in this study might be different from human gingival crevicular fluid.

### Assessment of the Antibacterial Activity of Dissolved Silver

The antimicrobial activity of silver in well documented. It has been found that a trace amount of silver can have a potential antibacterial activity. The minimum inhibition concentration (MIC) of nanosilver against oral pathogens can be as low as 0.05 mg/L [19]. In this study, 0.34 mg/L of silver was enough to provide an acceptable antibacterial activity. This result is supported by another study, which found that the release of silver to the external media from silver-coated dentine discs was 0.32 mg/L, an amount that caused the complete inhibition of growth of *S. mutans* [17]. A possible explanation for this inhibition is the direct contact toxicity of free silver ions against the bacterial cells [20]. Another possible explanation is the fact that free silver ions can penetrate bacterial cells and produce reactive oxygen species which cause oxidative stress and cell death [27]. This confirms the hypothesis that the dissolved silver from silver-coated titanium abutments can potentially prevent (or reduce) the incidence of pre-implant mucositis by releasing free silver ions into the gingival sulcus. Furthermore, it could be argued that this fraction of silver which has been released into the external media is safe for gingival fibroblast cells. Meran and co-workers found that around 1 mg/L of silver did not exhibit significant cytotoxicity against human fibroblast cells [28]. This confirms the fact that according to this study, if the same method of silver coating is applied to the dental implant abutment, silver release can inhibit bacterial growth in gingival sulcus without harming the gingival fibroblast cells; however, this might be affected by the release rate and bioavailability of the silver released into the gingival sulcus.

Regarding the antibiofilm (anti plaque) activity of the dissolved silver from silver-coated titanium dental implant abutments, the dissolved silver did not exhibit a significant antibiofilm activity as there was no significant difference in turbidity measurements between the control and treatment. Moreover, lactate production did not show a significant difference between the groups, indicating that the *S. mutans* in the control and treatment were metabolically active. The logical explanation for this finding is that bacterial cells in a biofilm are more protected when compared to suspended bacterial cells. This finding can also be explained by the definition of a biofilm itself as a “microbial-derived sessile community characterized by microbes that are irreversibly attached to a substratum and/or interface to each other, embedded in a matrix of extracellular polymeric substances produced by microbes”. The presence of extracellular polysaccharides in a biofilm can significantly protect bacteria against antimicrobials [29]. Since the biofilm is composed of organic molecules in which the bacterial cells are embedded, the bioavailability of silver ions in a biofilm will be significantly reduced as a result of the silver and -SH group affinity. Furthermore, the proteins in a biofilm can bind to the metal particles and form protein corona, which reduce the bioavailability and consequently inhibit the bactericidal activity of silver [30].

The antibiofilm activity of silver nanoparticles is well documented, for example, Besinis et al., 2014, found that nanosilver-coated dentine specimens significantly inhibited the formation of a biofilm by *S. mutans* on the coated surface [31]. However, their experimental condition was different from this study. In this study, the biofilm was already formed on the dish surface and then challenged with the dissolved silver, while Besinis and co-workers cultured the *S. mutans* on the silver-coated disc itself. Thus, this could support the fact that bacteria embedded in a biofilm can significantly be protected from metal particles. There are two scenarios to be discussed: first, whether silver can prevent biofilm formation, and second, whether silver can kill the bacterial cells in an already formed biofilm. In the current study, the dissolved silver could not inhibit the bacterial activity in a formed biofilm, but it showed significant antibacterial activity against suspended bacteria. The clinical significance of this finding is that the slow release of silver from a silver-coated titanium dental implant abutment into the gingival sulcus can significantly reduce the incidence of peri-implant mucositis.

## 4. Materials and Methods

### 4.1. Specimen Preparation

The titanium alloy selected for the current study is routinely used for constructing dental implants, and the method for preparing the discs was based on [9]. Briefly, grade five titanium alloy (Ti6Al4V) discs measuring 15 mm in diameter and 1 mm in thickness were prepared via laser cutting and then polished with sandpapers of different grid sizes. The discs were subsequently cleaned with an alkaline solution and 5% HCl as previously described [9]. The cleaned titanium discs were then coated with Ag NPs via an electroplating method. Briefly, the clean titanium discs were hung on a silver wire connected to the cathode of a voltage supply (BK Precision, 9174 DC power supply) while a fine silver sheet (1 mm thickness, 50 mm × 100 mm, Cooksongold Ltd., Birmingham, UK) comprised the anode. The discs and silver sheet were immersed in an electrolyte (0.2 M of AgNO_3_, 0.4 M of succinimide, and 0.5 M of KOH; all from Sigma Aldrich, UK) at 40 °C, and the voltage was adjusted to 1 V and left for 3 min. To increase the adhesion of silver nanoparticles to the substrate, the coated specimens were heated up to 500 °C for 1 h [25].

### 4.2. Surface Characterization of the Coating via EDS

Energy dispersive spectroscopy (EDS, spot 195 size, 10 μm; accelerating voltage, 15 kV; working distance, 10 mm) was used to assess the surface quality and quantity of the coated titanium discs as well as to carry out the surface metal analysis. For specimen preparation, a thin layer of chromium was sputter-coated onto the silver-coated specimens to increase conductivity. The data were analyzed using the Aztec 2.2 software supplied with the instrument.

### 4.3. Experimental Design

The experiment used 18 tubes divided into three groups: BHIB without *S. mutans* (Blank), grown *S. mutans* in BHIB (control), and *S. mutans* exposed to conditioned BHIB containing released silver from silver-coated titanium discs (treatment) (*n* = 6). All tubes were incubated for 24 h at 37 °C in a microaerophilic condition using a candle-jar technique. The antibacterial effect of the dissolved silver on *S. mutans* was then investigated by measuring the turbidity and antibiofilm effect as well as carrying out a lactate production assay to assess the bacterial metabolic activity.

### 4.4. Silver Release

In order to release the silver ions from the surfaces of the silver-coated titanium implant discs, the discs were immersed in 1.8 mL of brain heart infusion broth (BHIB) and incubated at 37 °C for 24 h (Figure 7). To measure the silver release, 0.4 mL of silver-conditioned broth was acidified with 20 µL of 70% nitric acid, and the Ag^+2^ content of the sample was determined via inductively coupled plasma mass spectrometry ICP-MS (ICP-MS, X Series 2, Thermo Scientific, Hemel Hempstead, UK) against matrix-matched standards.

### 4.5. Isolation and Identification of S. mutans

The bacterial isolate *S. mutans* used for this study was isolated from a human dental plaque. The plaque sample was taken from a carious lesion using the tip of a sterilized toothpick. The sample was then vortexed with 1 mL of sterile, normal saline for one minute to disperse the plaque and obtain a homogeneous suspension. The sample was diluted with sterile saline and plated on blood agar and Mitis Salivarius agar (MSA) (Himedia, India), which were supplemented with a 1% potassium tellurite solution. The sample was incubated at 37 °C for 48 h in a microaerophilic condition using a candle-jar technique. After the incubation, the colonies were morphologically checked and biochemically tested. Granular colonies with frosted-glass appearances exhibiting either alpha or gamma hemolysis patterns on the blood agar were identified as *S. mutans*.

### 4.6. Preparation of Bacterial Suspensions

A single colony of *S. mutans* was inoculated in 5 mL of BHIB broth in a screw-capped tube and incubated at 37 °C for 48 h in under microaerophilic conditions, using the candle-jar technique. Then, a concentration of 1.5 × 108 CFU/mL was prepared.

### 4.7. Preparation of Treatment and Control Tubes

To set up the experiment, six tubes (treatment tubes), each containing 0.6 mL of silver-conditioned broth with 0.6 mL of bacterial suspension, and six control tubes, each containing 0.6 mL of bacterial suspension and 0.6 mL of sterile broth, were arranged.

### 4.8. Assessing the Antibacterial Activity

The turbidity method was used to determine the effect of the dissolved silver on bacterial growth. Aliquots of 200 μL were withdrawn from the treatment and control tubes and added to a 96-well plate (flat-bottom with lid). The optical density was then measured using a plate reader (Bio-Tek ELX800 Microplate Reader) at 595 nm.

### 4.9. Assessing the Antibiofilm Activity

The tube method was performed as described by [32]. The control and treatment tubes’ contents were discarded, and the tubes were washed gently with Milli-Q water to remove non-adherent bacteria. For the biofilm staining, 1.2 mL of crystal violet (0.1%) was added to each tube and left at room temperature for 30 min, after which the stain was discarded. The washing step was repeated, and the tubes were left to dry in an inverted position at room temperature. The biofilm formation was detected via the presence of a visible film on the wall and bottom of the tube. To remove the biofilm, 3 mL of ethanol was added to each tube, and then the detached biofilm in ethanol was withdrawn. After that, 200 μL of each sample was added to a 96-well plate (flat-bottom with lid), and the optical density was measured using a plate reader (Bio-Tek ELX800 Microplate Reader) at 595 nm.

### 4.10. Determination of the Lactate Production

Since *S. mutans* are anaerobic bacteria, the appearance of lactate in BHIB is an indication of the presence of metabolically active bacterial cells. After 24 h of exposure to the dissolved silver, 70 mL of the remaining volume from each well was transferred to new V-bottom 96-well microplates and then centrifuged at 2000 rpm for 10 min to pallet the bacteria. After that, 10 µL of the supernatant from each well was removed to new flat-bottom 96-well plates for the lactate assay, according to [33]. The assay used 10 µL of the sample and 211 µL of assay reagents consisting of 200 mL of a solution containing 0.4 M of hydrazine and 0.5 M of glycine, 10 mL of 40 mM of nicotinamide adenine dinucleotide (NAD+, Melford Laboratories Ltd., Suffolk, UK), and 1 mL of 1000 units/mL of lactate dehydrogenase. The reagents and the sample were then incubated for 2 h, and the absorbance was read at 340 nm via a plate reader (Bio-Tek ELX800 Microplate Reader).

### 4.11. Statistical Analysis

Data are presented as means ± S.E.Ms and were analyzed using Statgraphics, version 16. Standard skewness and standard kurtosis were used to determine whether the data were normally distributed. To locate the significant differences between the groups, data were subjected to a one-way ANOVA, followed by Tukey’s test. All statistical analyses used a 95% confidence limit, and *p* values < 0.05 were considered statistically significant.

## 5. Conclusions

The silver released into the BHIB was found to inhibit bacterial growth in a suspension. However, the antibiofilm efficacy of the silver was not significantly reported. The clinical significance of this finding is that if the titanium dental implant abutment is coated with silver, the silver released (dissolved) from the coating into the gingival crevicular fluid (gingival sulcus) can significantly reduce the incidence of peri-implant mucositis, which consequently reduces the incidence of peri-implantitis, thus increasing the clinical success rate of the dental implant treatment.

## 6. Limitations and Future Work

The limitation of this study is that the dissolution of silver in BHIB might be different from gingival crevicular fluid as their biological contents might be different; thus, the antibacterial property of silver dissolved from the coated abutment into the gingival sulcus will be affected. It could be recommended to conduct a study investigating the amount and antibacterial activity of the dissolved silver in gingival crevicular fluid.

## Figures and Tables

**Figure 1 antibiotics-12-01097-f001:**
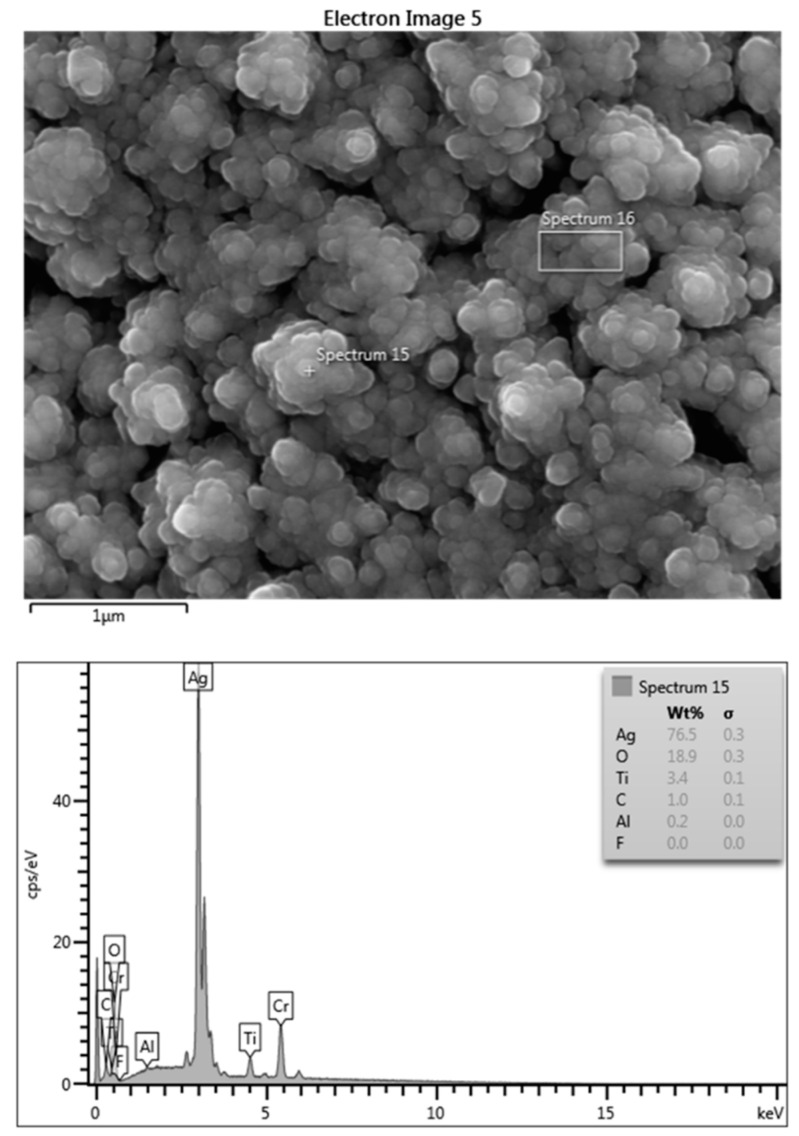
Surface morphology of silver-coated titanium disc; note the nano-sized particles and silver peak indicating that most of the surface is covered by silver.

**Figure 2 antibiotics-12-01097-f002:**
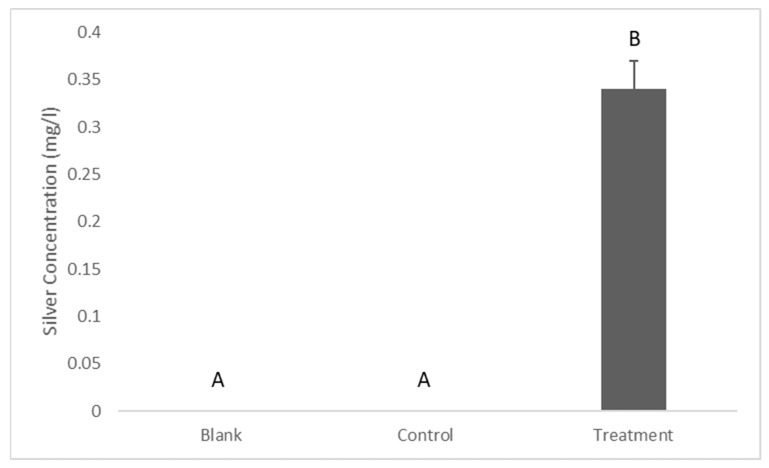
Silver release from the silver-coated titanium implant discs to the BHIB. Data are shown as means ± S.E.Ms, different letters indicate significant differences between the groups. One-way ANOVA; *p* ≤ 0.05.

**Figure 3 antibiotics-12-01097-f003:**
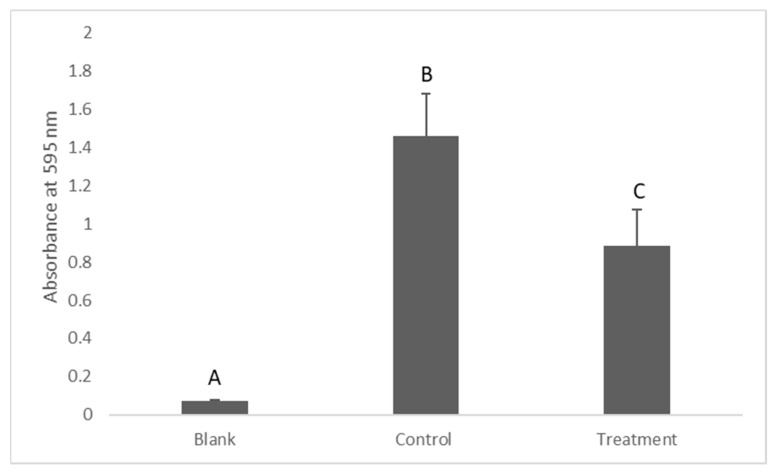
Turbidity measurement to assess an antibacterial activity of dissolved silver. Data are shown as means ± S.E.Ms. Different letters indicate significant difference between the groups. One-way ANOVA; *p* ≤ 0.05.

**Figure 4 antibiotics-12-01097-f004:**
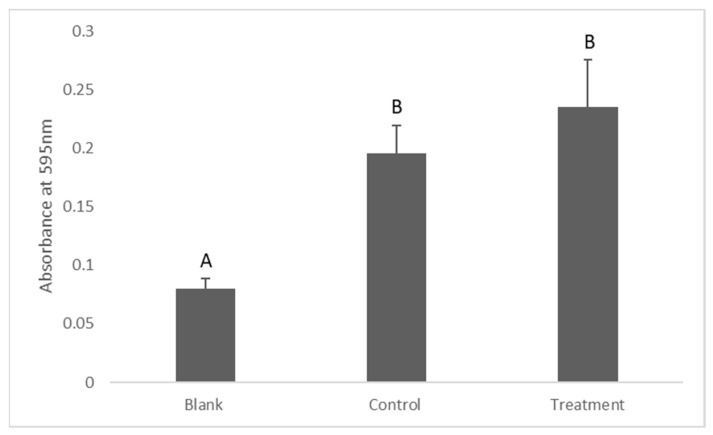
Turbidity measurement to assess the effect of dissolved silver on the biofilm formed by *S. mutans*. Data are shown as means ± S.E.Ms. Different letters indicate significant differences between the groups. One-way ANOVA; *p* ≤ 0.05.

**Figure 5 antibiotics-12-01097-f005:**
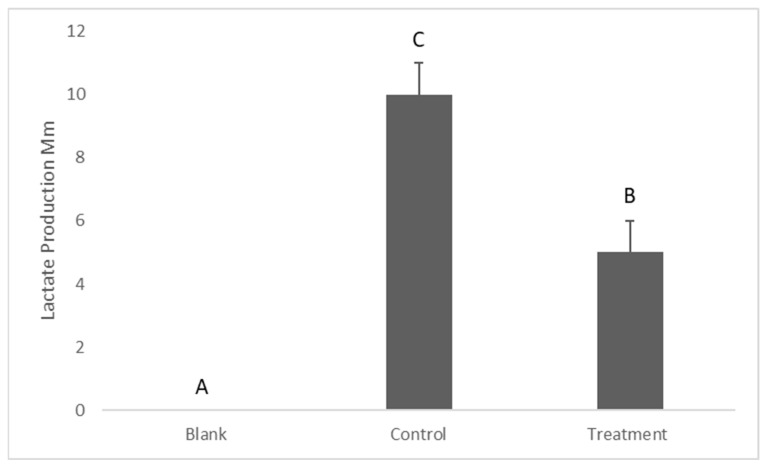
Determination of lactate production by *Streptococcus mutans*. Data are shown as means ± S.E.Ms, different letters indicate significant difference. One-way ANOVA; *p* ≤ 0.05.

**Figure 6 antibiotics-12-01097-f006:**
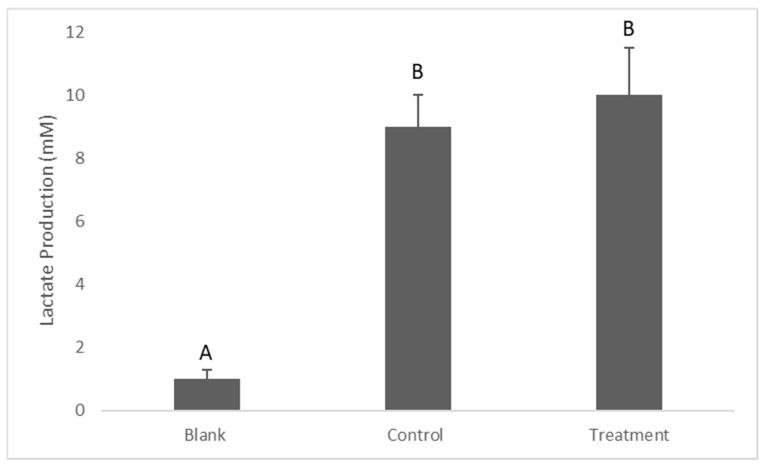
Determination of lactate production by *Streptococcus mutans*. Data are shown as means ± S.E.Ms, different letters indicate significant differences. One-way ANOVA; *p* ≤ 0.05.

**Figure 7 antibiotics-12-01097-f007:**
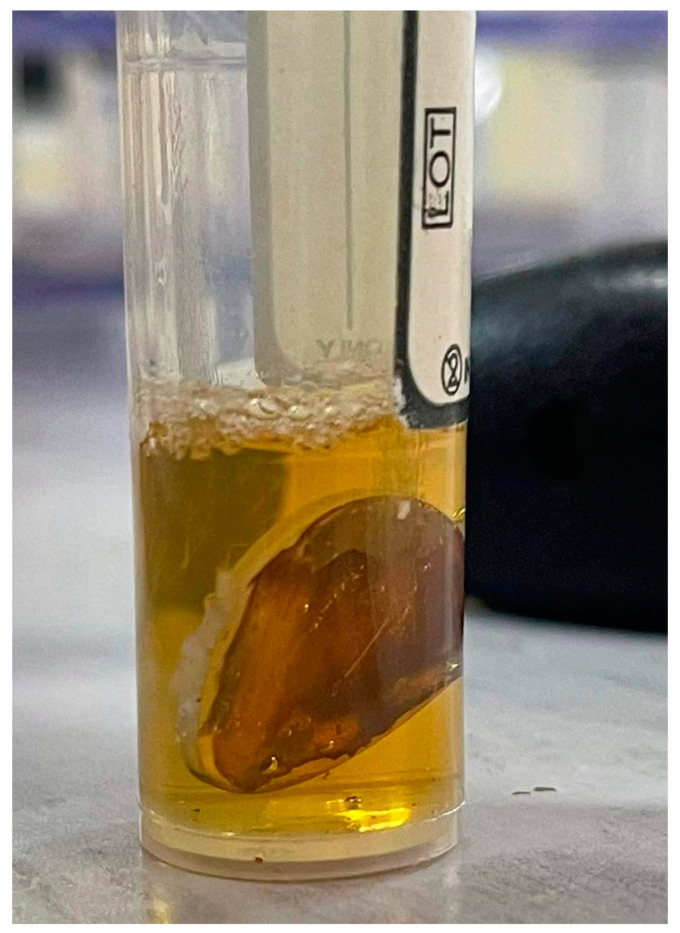
Silver-coated titanium implant disc immersed in BHIH for silver release (dissolution).

## Data Availability

Not applicable.

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
