# Peer review of "Antibacterial Activity of Dissolved Silver Fractions Released from Silver-Coated Titanium Dental Implant Abutments: A Study on Streptococcus mutans Biofilm Formation"

_antibiotics, 2023, doi:10.3390/antibiotics12071097_

Round 1

Reviewer 1 Report

The manuscript presents a comprehensive research study that delves into the antibacterial properties of dissolved silver derived from silver-coated titanium implants, specifically targeting Streptococcus mutans. The work is well-defined, with the authors clearly articulating a concise and straightforward hypothesis. The methodology employed in this study is appropriate. The discussion and results sections are well presented, offering a clear and concise analysis of the findings.  

Nevertheless, it is of very  important that the authors provide a detailed explanation and in-depth discussion regarding the utilization of Brain Heart Infusion Broth instead of an oral medium solution. Opting for an oral medium solution would offer an environment that closely mimics the conditions found in the oral cavity. Furthermore, it would be beneficial for the authors to address the possibility of comparing the release of silver ions in both mediums. Moreover, it is essential to evaluate the silver ion release from the silver-coated titanium implant discs in relation to the minimum inhibitory concentration (MIC) of silver against Streptococcus mutans, as reported in various scientific studies. It is worth noting that this value can vary depending on factors such as the specific strain of S. mutans, the experimental conditions employed, and the form of silver being tested (e.g., silver ions, silver nanoparticles, etc.). However, the findings presented in this manuscript (0.3 mg/l) indicate a significantly lower level of silver ions compared to the average values reported previously.

Author Response

Dear Reviewer,
Thank you very much for your interesting and productive comments regarding the manuscript.
1- About using oral medium instead of BHIB, I have added sentences in the introduction 
and discussion that silver dissolution in biological media depends on many factors like 
content and volume of the media, duration of exposure, coating procedure, silver shape 
and size …etc
I understand that silver dissolution and bioavailability varies in different biological media 
and BHIB is not exactly the same as gingival crevicular fluid. However, providing an 
optimum growth condition for the bacteria in-vitro is critical. In addition, we need at 
least 20 ml of growth medium for this study, collecting this amount of gingival crevicular 
fluid is very challenging. Moreover, most of the articles that have investigated 
antibacterial activity of biomaterials in-vitro, have used BHIB (specially with S.Mutans). 
for example, 
-Besinis, A., De Peralta, T. and Handy, R.D., 2014. The antibacterial effects of silver, 
titanium dioxide and silica dioxide nanoparticles compared to the dental disinfectant 
chlorhexidine on Streptococcus mutans using a suite of bioassays. Nanotoxicology, 8(1), 
pp.1-16.
-Gunputh, U.F., Le, H., Lawton, K., Besinis, A., Tredwin, C. and Handy, R.D., 2020. 
Antibacterial properties of silver nanoparticles grown in situ and anchored to titanium 
dioxide nanotubes on titanium implant against Staphylococcus 
aureus. Nanotoxicology, 14(1), pp.97-110.
-Besinis, A., De Peralta, T. and Handy, R.D., 2014. Inhibition of biofilm formation and 
antibacterial properties of a silver nano-coating on human dentine. Nanotoxicology, 8(7), 
pp.745-754.
Regarding comparing silver release in BHIB and oral medium which a critical point, I 
added this sentence in the future work section
2- Regarding the MIC, evaluating the silver ion release in our study in relation to the MIC of 
silver against S.Mutans. Sivler dissolution in our study is higher than the reported MIC in 
other studies. Some studies have showed that the MIC of silver against S.Mutans is 0.05 
mg/l (Pokrowiecki et al., 2017), other study have found MIC of dissolved silver against 
S.Mutans to by 0.09 mg/l. 
I added this point to the discussion.
Regarding the factors affecting the bioavailability and MIC of silver, I have added these 
points to the introduction.
Kind Regards 
Corresponding Author

Reviewer 2 Report

Title: Antibacterial Activity of Silver-Coated Titanium Dental Implant Abutments: A Study on Streptococcus mutans Biofilm Formation

Abstract: The abstract provides a concise overview of the study's objectives, methods, results, and conclusions. However, it could be enhanced by including specific quantitative results and highlighting the significance of the findings. Additionally, it would be helpful to mention the potential clinical implications of the research.

Introduction: The introduction provides a good background on the use of titanium dental implant abutments and the challenges associated with bacterial biofilm formation. To improve the introduction, consider incorporating more recent literature to support the importance of the topic. E.g. Haugen, H.J.; Makhtari, S.; Ahmadi, S.; Hussain, B. The Antibacterial and Cytotoxic Effects of Silver Nanoparticles Coated Titanium Implants: A Narrative Review. Materials 2022, 15, 5025. https://doi.org/10.3390/ma15145025

Vilarrasa, J.; Delgado, L.M.; Galofre, M.; Alvarez, G.; Violant, D.; Manero, J.M.; Blanc, V.; Gil, F.J.; Nart, J. In vitro evaluation of a multispecies oral biofilm over antibacterial coated titanium surfaces. J. Mater. Sci. Mater. Med. 2018, 29, 164.

Salaie, R.N.; Besinis, A.; Le, H.; Tredwin, C.; Handy, R.D. The biocompatibility of silver and nanohydroxyapatite coatings on titanium dental implants with human primary osteoblast cells. Mater. Sci. Eng. C Mater. Biol. Appl. 2020, 107, 110210

Pokrowiecki, R.; Zareba, T.; Szaraniec, B.; Palka, K.; Mielczarek, A.; Menaszek, E.; Tyski, S. In vitro studies of nanosilver-doped titanium implants for oral and maxillofacial surgery. Int. J. Nanomed. 2017, 12, 4285–4297

Additionally, providing a clear statement of the research objectives and hypothesis would enhance the clarity of the introduction.

Materials and Methods: The materials and methods section provides a detailed description of the experimental procedures and techniques employed. However, it would be beneficial to include additional information on the specific strains of Streptococcus mutans used, as well as details on the growth conditions and duration. Additionally, specifying the statistical analysis used would improve the transparency of the study.

Results: The results section presents the findings of the study, including the antibacterial and antibiofilm activities of the silver-coated titanium abutments. To improve the clarity of the results, consider organizing the information into subheadings and utilizing tables or figures to present the data visually. Additionally, including statistical analyses and p-values where applicable would enhance the rigor of the results section.

Discussion: The discussion provides a comprehensive interpretation of the results in light of the study objectives. To strengthen the discussion, consider comparing the findings with previous studies in the field and discussing potential mechanisms underlying the observed effects. Additionally, address any limitations of the study and suggest avenues for future research.

Conclusion:

The conclusion briefly summarizes the key findings of the study. However, it would be beneficial to expand on the clinical implications of the results and discuss the potential impact on reducing the incidence of peri-implant mucositis. Consider rephrasing the conclusion to emphasize the practical significance of the study's findings.

General Suggestions:

Ensure consistent formatting and citation style throughout the manuscript.

Review the manuscript for any grammatical errors or awkward sentence structures.

Provide more specific details on the methodology and experimental design to facilitate replication of the study.

Consider including visual aids such as figures or tables to present data more effectively.

Address any limitations of the study and provide suggestions for future research directions.

Enhance the clarity of the manuscript by using subheadings and logical organization of the content.

Overall, the manuscript provides valuable insights into the antibacterial activity of silver-coated titanium dental implant abutments. With the suggested improvements, the manuscript will be better positioned for publication in the Antibiotics journal.

The language used in this manuscript is generally academic and formal, which is appropriate for a scientific research paper. The sentences are structured well, and the vocabulary is technical and specific to the field of study. However, there are a few areas where language improvements could be made:

Clarity and Precision: Some sentences could be rephrased or simplified to enhance clarity and precision of the information presented. This could involve breaking down complex ideas into smaller, more digestible units or using more straightforward language when possible.

Sentence Structure: While the majority of the sentences are well-constructed, there are instances where sentence structure could be improved for better flow and readability. Reviewing the manuscript for sentence variety and ensuring that sentences are not overly long or convoluted will enhance readability.

Technical Terminology: Given the scientific nature of the manuscript, technical terminology is necessary. However, it is important to strike a balance between technical language and accessibility. When introducing complex terms, consider providing brief explanations or definitions to aid readers who may not be familiar with the specific terminology.

Consistency: Ensure consistency in terms of tense, verb usage, and formatting throughout the manuscript. This will help maintain a cohesive narrative and improve overall readability.

Grammar and Proofreading: Thorough proofreading is essential to catch any grammatical errors or typos that might be present. Pay attention to subject-verb agreement, proper use of punctuation, and correct citation formatting.

By addressing these language aspects, the manuscript's readability, clarity, and overall quality can be enhanced.

Author Response

Dear Reviewer,
Thank you very much for your interesting and productive comments regarding the manuscript.
Regarding the title, I have modified it according to your kind suggestion. However, it might be 
useful to keep (Dissolved Silver Fractions Released from Silver-Coated Titanium Dental Implant 
Abutments) the highlighted words in the title as this study was conducted on the released 
(dissolved) silver. What would you kindly suggest? 
1- Abstract: I added quantitative results and highlighted the clinical significance of the 
study
2- Introduction: thanks for proposing these important articles! I added them to the 
introduction. I also added specific objectives at the end of introduction
3- Materials and methods: the strain used in our study was S.Mutans isolated from dental 
caries and identified biochemically. They were incubated fro 48 hours at 37 degree in 
microaerophilic using candle-jar technique. I added these details to the methods 
section. Regarding the statistics, the data were normally distributed, so we followed 
one- way ANOVA using Tukey test. (also added to the methods section)
4- Results: I have subdivided the results section into 3 section , a- confirming silver release 
to the media by ICP-MS b- assessment of the antibacterial activity c-assessment of the 
antibiofilm activity d- lactate production. Figures of each finding is included. Regarding 
the p values, I added the actual p values where relevant .
5- Discussion: I added few articles to compare our findings with (eg; Gunputh et al 2020 
and besinis et al 2014 for antibiofilm activity). I have also added limitations and future 
work as a new section after conclusion.
6- Conclusion: I rephrased the conclusion according to your kind suggestions, pointing to 
the clinical significance of the findings.
7- I have added a new section (limitations and future work) according to your kind 
suggesting.
According to general suggestions: 
1- Consistent formatting was corrected, there were some mistakes regarding consistency, I 
corrected them 
2- I tried to thoroughly proofread the manuscript and catch grammatical errors.
3- Experimental design was expanded to facilitate replication of the study
4- Limitation and future work were added as a new section
5- A subheading was added in a discussion to improve the flow and clarity 
Regarding the language proofreading, thorough proofreading was made and many typos, 
inconstancies, awkward sentence structures were found and corrected (track changed in the 
manuscript).
Kind Regards
Corresponding Author

Round 2

Reviewer 1 Report

After careful consideration, I find no further comments to provide regarding the manuscript. In my opinion,  the manuscript is ready for acceptance.

Reviewer 2 Report

no further comments!